# Professionals' perspectives on factors affecting GP trainees' patient mix: results from an interview and focus group study among professionals working in Dutch general practice

Sarah de Bever  ,[1] Suzanne C van Rhijn,[1] Nynke van Dijk,[1] Anneke Kramer,[2] Mechteld R M Visser[1]

[1]Department of General Practice/ GP Specialty Training Program, Amsterdam UMC, University of Amsterdam, Amsterdam School of Public Health, Amsterdam, The Netherlands
[2]Department of Public Health and Primary Care, Leiden University Medical Center, Leiden, The Netherlands

**Correspondence to**
Sarah de Bever;
s.debever@amsterdamumc.nl

## ABSTRACT

**Introduction** Seeing and treating patients in daily practice forms the basis of general practitioner (GP) training. However, the types of patients seen by GP trainees do not always match trainees' educational needs. Knowledge about factors that shape the mix of patient types is limited, especially with regard to the role of the professionals who work in the GP practice.

**Aim** We investigated factors affecting the mix of patients seen by GP trainees from the perspective of professionals.

**Design and setting** This qualitative study involved GP trainees, GP supervisors, medical receptionists and nurse practitioners affiliated with a GP Specialty Training Institute in the Netherlands.

**Methods** Twelve focus groups and seven interviews with 73 participants were held. Data collection and analysis were iterative, using thematic analysis with a constant comparison methodology.

**Results** The characteristics of patients' health problems and the bond between the doctor and patient are important determinants of GP trainees' patient mix. Because trainees have not yet developed bonds with patients, they are less likely to see patients with complex health problems. However, trainees can deliberately influence their patient mix by paying purposeful attention to bonding with patients and by gaining professional trust through focused engagement with their colleagues.

**Conclusion** Trainees' patient mix is affected by various factors. Trainees and team members can take steps to ensure that this mix matches trainees' educational needs, but their success depends on the interaction between trainees' behaviour, the attitudes of team members and the context. The findings show how the mix of patients seen by trainees can be influenced to become more trainee centred and learning oriented.

## INTRODUCTION

Seeing and treating patients form the basis of training for medical graduates specialising to become general practitioners (GPs).[1 2] Thus, their development of competence and skills depends on the mix of patients seen.

### Strengths and limitations of this study

► We conducted focus groups and interviews with 73 participants who together represented all professional roles in general practice (general practitioner (GP), medical receptionist, nurse practitioner, GP trainee).
► Qualitative analysis allowed for an in-depth exploration of factors affecting trainees' patient mix.
► Combining all these perspectives, this study provides a rich insight into factors that affect trainees' patient mix.
► While patients' perspectives are missing from this study, literature on their perspectives suggest similar findings.
► The results are most applicable to settings in which continuity of care is important.

According to Ericsson's theory of deliberate practice, mastery of competencies is achieved through repetitive practice, monitoring and timely feedback from a trusted coach followed by further practice.[3 4] In the clinical workplace setting, this means repetitive cycles of the same patient symptoms or situations. Yet, in everyday practice, the mix of patients seen by GP trainees cannot always be organised in this way.[5] The mix seems to be affected by a number of factors other than the educational needs of the trainee.[6 7] To make GP training more effective, a thorough understanding of all factors affecting the patient mix of GP trainees is needed.

Previous studies of trainees' patient mix mostly focused on patients' willingness to consult the trainee.[8–11] However, trainees, their supervisors and the practice environment probably also affect the patient mix. It has been found, for example, that GP trainees influence their patient mix in order to fill their knowledge

gaps,[12] but this requires developmental space.[2 13] Supervisors can provide this space, however they need to trust the trainee first.[14–16] The relationship between GP trainee and supervisor affects the trainees' ability to steer his/her learning, and therefore probably also their patient mix.[2 12] Contextual factors, such as the work climate, may also influence which patients are seen by the trainee. While trainee workplaces should 'invite and support' learning,[17–19] some are more work oriented than supportive for learners.[6 20] For example, medical receptionists are responsible for allocating patients, and they may not necessarily prioritise trainees' educational needs when allocating patients. Many studies have investigated how trainees learn,[1 12 21 22] how supervisors develop their trust,[23 24] the importance of work relationships and the influence of context on learning,[2 12 19] but none have looked at the impact of the trainees' patient mix. Yet, this information is needed in order to support training in the general practice.

In this study, we investigated factors influencing the patient mix of GP trainees. To this end, we carried out a qualitative study involving different health professionals (GP trainees, their supervisors, medical receptionists and nurse practitioners) in a general practice setting. Findings can be used to ensure that trainees see a mix of patients appropriate to their needs, thereby optimising workplace-based learning.

## METHODS
### Design
Focus group meetings and additional semi-structured interviews were used in this qualitative study. Focus groups were chosen because they provide the opportunity for in-depth investigation of a topic through group discussion.[25]

### Participants
GP trainees, GP supervisors, medical receptionists and nurse practitioners affiliated with the GP Specialty Training Institute of the Amsterdam UMC (University Medical Centers), University of Amsterdam, were invited to participate. All participants came from different general practices. See table 1 for more details regarding the number of participants; recruitment methods used; the number, place and duration of the focus groups or interviews. After the initial analysis, we held six interviews with trainees to deepen our first impressions. We conducted purposive sampling on final year trainees because they appeared to be the most informative participants. While all focus groups were homogenous with regard to profession we ensured that they were heterogeneous with regard to practice organisation (single practice, group practice or health centre) and level of the trainee to stimulate discussion.

### Setting
In the Netherlands, all residents are registered with a general practice. GPs are generalists, gatekeepers for specialist and hospital care, and patients' first contact with the healthcare system.[26] Most general practices are in the community, so there is often a strong, long-standing relationship between GPs and patients.[26] Personal care and continuity of care are core values in Dutch general practice. GPs can work alone (solo practice), with two or more GPs (group practice), or be organised in larger health centres with several GPs

| Table 1 | Recruitment methods and number, place and duration of the focus groups or interviews | | | | |
|---|---|---|---|---|---|
| | **Design** | **Trainees (Y1/Y2/Y3)** | **Supervisors** | **Medical receptionist** | **Nurse practitioners** |
| Recruitment | Focus group | Multiple presentations during the educational programme in 2016 and 2017 | Email prior to training day: June 2016 and presentation on training: day April 2018 | Presentation during training day: March 2016 | Email prior to training day: April 2016 |
| | Interview | By teachers of educational programme | NA | Presentation during training day: March 2016 | NA |
| Location/time | Focus group | Regular educational programme day at the department between May and June 2016 | Training day for supervisors in June 2016[2] and April 2018[1] | After work hours at GP department | Training day April 2016 |
| | Interview | Trainees' place of preference between January and February 2017 | NA | MRs' place of preference in December 2016* | NA |
| Duration | Focus group | 40 min | 60 min | 60 min | 60 min |
| | Interview | 20 min | NA | 30 min | NA |

*For logistic reasons, one medical receptionist was interviewed separately.
GP, general practitioner; MR, Medical receptionist; NA, Not applicable.

de Bever S, et al. BMJ Open 2019;**9**:e032182. doi:10.1136/bmjopen-2019-032182

and other healthcare providers. In their practice, GPs are supported by one or more medical receptionists and nurse practitioners. In general, medical receptionists are responsible for allocating patients to specific doctors, while nurse practitioners are responsible for the care of patients with chronic diseases, elderly patients or patients with psychosocial problems.

In the Netherlands, the three-year postgraduate GP training programme is offered by eight departments of general practice, all affiliated with academic medical centres. In their first and third years, trainees work in a general practice fourdays each week. On the fifth day, trainees follow an educational programme at one of the departments. In the general practice, each GP trainee is supervised and coached by an experienced GP (supervisor). Most of the time, trainees work independently and discuss their patients on a regular basis, or immediately if needed, with their designated supervisor. In their second year, trainees rotate through hospital, nursing home and psychiatric care settings. GP supervisors, nurse practitioners and medical receptionists affiliated with the GP Specialty Training Institute of the Amsterdam UMC(University of Amsterdam) are offered training days. Training days for GP supervisors are held multiple times a year and are compulsory. Training days for nurse practitioners and medical receptionists are held twice a year and are voluntary.

### Patient and public involvement
We did not involve patients nor the public in this study. The research question was based on previous studies and was part of a larger, peer-reviewed study protocol. The results will be disseminated through training programme for GP supervisors and trainees. We developed workshops in which supervisors and trainees are informed about the main results of our study and are taught skills to influence their patient mix. These workshops are part of further investigations on trainees' patient mix and are implemented within the GP specialty training programme of theAmsterdam UMC (University of Amsterdam).

### Data collection
Topic lists, covering the role of trust, supervision, patients' perspectives and preferences, continuity of care and contextual factors (eg, practice size and number of employees), were used for the focus meetings and interviews. Topic lists were continuously updated based on findings. Focus groups were led by skilled moderators experienced in qualitative research and familiar with the scope of this study. The first author (SdB) observed most of the focus groups. If this was not possible for logistic or ethical reasons, another skilled observer (SvR) familiar with the study protocol took over. All focus groups and interviews were audio-taped and transcribed verbatim. Transcripts were pseudonymised.

### Data analysis
We used thematic analysis with a constant comparison methodology to guide our analysis.[27] Initially, two team members (SdB and SvR) independently coded two transcripts. After these first two transcripts, SdB and SvR discussed the initial coding scheme until they reached consensus. Next, five more transcripts were independently coded by SdB and SvR, using constant comparison with the earlier codes and adding new codes were needed. These codes were discussed in detail between SdB, SvR and MV until consensus was reached. Data from subsequent transcripts were further coded by SdB using the discussed codes scheme and, where needed, new codes were added and other code were merged or adopted. During this phase, SdB, SvR and MV met on regular basis to discuss the code scheme and overarching themes. To fully explore the themes and their mutual relationship, multiple diagrams and figures were created. After proximally 13 transcripts the code scheme seemed sufficient to code the last transcripts, indicating data saturation. This final code scheme and the earlier figures were discussed within the whole research group. The research team included a GP trainee/PhD student (SdB) and four researchers, all experienced in medical education research and qualitative methodology: an educationalist and medical doctor (NvD), a cognitive psychologist (MV), a GP (AK) and a research assistant/educationalist (SvR).

Prior to each focus group or interview, participants were given study information and informed consent was obtained.

## RESULTS
In total, eleven focus groups were held, of which five with GP trainees (one with first-year trainees, two with second-year trainees, two with third-year trainees), three with supervisors, two with nurse practitioners and two with medical receptionists. Additionally, six interviews were held with third-year trainees before data saturation was reached. In total, we included 73 participants. The participants' demographics can be found in table 2.

Our analysis revealed three major themes that together explain which factors influence GP trainees' patient mix: *Disease characteristics & doctor-patient relationship, Overcoming the lack of doctor-patient relationship* and *Influencing factors*. Online supplementary appendix I provides an overview and summary of each theme and their subthemes. The emerged themes are closely interrelated with each other. For a thorough understanding of our findings, an understanding of these relationship is needed. Therefore, we will first start with a summary of our findings and how they relate to each other, before explaining the themes and their subthemes separately.

### Summary
The allocation of patients to either a GP trainee or GP supervisor, and therefore trainees' patient mix, is influenced by several interacting factors. The relationship between doctor (supervisor or trainee) and patient in the context of the patients' health problem played an

**Table 2** Demographic data

| | Trainees (n=37) | Supervisors (n=18) | Medical receptionists (n=6) | Nurse practitioners (n=12) |
|---|---|---|---|---|
| Age, years (median, range) | 30 (27–35) | 56 (35–62) | 53 (39–60) | 53 (38–63) |
| Gender (male/female) | 6 (16.2%) /31 (83.8%) | 5 (27.8%) / 13 (72.2%) | 0/6 (100%) | 0/12 (100%) |
| Practice organisation | | | | |
| Single | 12 (32.4%) | 5 (27.8%) | 0 | 1 (8.3%) |
| Group | 16 (43.2%) | 10 (55.6%) | 5 (83.3%) | 7 (58.3%) |
| Healthcare centre | 3 (21.6%) | 3 (16.7%) | 1 (16.6%) | 4 (33.3%) |
| Level of training (Y1/Y2/Y3)* | 4 (10.8%) / 16 (43.2%) / 17 (45.9%) | NA | NA | NA |
| Years as professional† (median, range) | NA | 23.5 (7–33) | 15 (5–43) | 14 (7–17) |
| N of focus groups/ interviews | 5/6 | 3/0 | 2/1 | 2/0 |
| Total participants in focus groups (range in each focus group) | 31 (5–8) | 18 (5–7) | 5 (2–3) | 12 (6) |

*Only applicable for trainees.
†Only applicable for supervisors, MR and NP.
NA, Not applicable.

essential role. Trainees obviously did not have a trustful relationship with patients yet, but both they and team members could compensate for this in various ways. Whether this occurs in practice depended on three influencing factors: trainees' behaviour, attitudes of the team and context (figure 1). Below we will first present how the doctor-patient relationship affects trainees' patient mix. Subsequently, we will explain how practices may overcome the lack in trusting relationship between trainee and patient, and how this process is influenced by trainees' behaviour, attitudes of the team and the context (influencing factors).

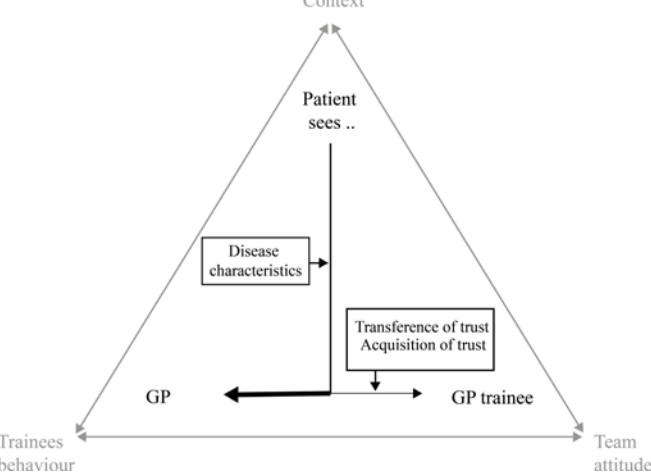

**Figure 1** Factors influencing the patient mix of general practitioner trainees.

### Disease characteristics and doctor-patient relationship

According to our participants, a central factor affecting the patient mix of trainees was the doctor–patient relationship in relation to the patients' health problem. The interviewees noticed that patients generally preferred to see their 'own' doctor because the doctor knew the patient and his/her (disease) history, social context and was trusted. However, patients' preference could change depending on the perceived severity of their health problems.

*NP no. 15: If someone comes with, for instance, heart rhythm disturbances or chest pain, they are happy if they can be seen quickly and that can also be a trainee. But if they come with mental health problems, and their GP knows the family situation etc, then they prefer to see that doctor, because he/she knows about them and their situation.*

In general, our participants expressed it was felt that trainees were not long enough in a practice to really become the trusted doctor. This lack of a trainee–patient relationship seemed to affect the patient mix, especially when patients came with chronic and/or complex conditions.

*Trainee no. 1: Yes, I think that, this year, that I have had relatively little experience with chronic health problems and their treatment. If my supervisor is away a week, then it gets interesting. Otherwise they are mainly seen by my supervisor.*

### Overcoming the lack of doctor–patient relationship
#### Transference of trust

Findings showed that team members trusted by patients, such as the medical receptionist and the supervisor, could transfer this trust to the trainee by emphasising the

de Bever S, *et al. BMJ Open* 2019;**9**:e032182. doi:10.1136/bmjopen-2019-032182

trainees' benefits. For example, trainees' previous work experience, pointing out that the trainee would have more time for them, and might have a new take on their problems. Medical receptionists in particular used these arguments if a patient was in doubt about who to see:

*MR no. 3: But sometimes if they've [the patient] had a problem for a long time, I'll say " Who knows, give it [seeing the trainee] a go, it may shed new light on the problem. And then they say "OK, I'll try it".*

This process could be strengthened by the presence of the supervisor or nurse practitioner during the first consultation of the trainee with the patient. If all went well, the patient could be transferred to the care of the trainee:

*Trainee no. 2: I had a patient who needed palliative care in the first 2 weeks of the year and then my supervisor said to the patient, when we were at his bedside "Look, I trust [name of trainee], it's OK. I have to go to a conference; he'll take over". I hardly knew the patient but he said " OK, we'll do that" And everything went OK. But because he [the supervisor] made it explicit at that moment, it took away a lot of worry and stress.*

### Acquisition of trust
Although trainees obviously did not immediately have a trustful doctor–patient relationship, this could develop rapidly, often after a successful first encounter.

*MR no. 1: But if they [patients] have seen [a trainee] once, perhaps because their 'own' doctor couldn't see them, and that goes well, then there is no problem with subsequent appointments.*

Whether the trainee gained the patients' trust and developed a good doctor-patient relationship depended to a large extent on the trainees' behaviour (see 'Trainees' behaviour').

### Influencing factors
#### Trainees' behaviour
Trainees could affect their patient mix directly or indirectly. Directly through investing time and energy in building a relationship with their patients, through which trainees could become the preferred doctor.

*Trainee no. 2: I had a female patient with a BIRADS-4 [Breast Imaging Reporting and Data System (BIRADS). A standardized reporting system for mammography, BIRADS 4 indicates that a suspicious node has been seen] and I had to tell her. And that was population screening. She started treatment. I phoned her every 2 weeks. She said "I appreciate you phoning to ask how things are going". I try to do that sort of thing. Yes, I think you can gain people's trust in that way.*

Trainees could influence their patient mix more indirectly by creating a supportive work environment. As stated earlier, trainees are dependent on other team

members when it comes to patient allocation or transfer. However, these team members first need to feel that they can trust the trainee before they will allocate patients to the trainee. Trainees could influence this in several ways, one of which was being keen, as explained by the following medical receptionist who was asked what trainees can do to speed up trust:

*MR no. 3: Well, by mentioning it–if you have patients with this or that problem, let me see them. Some trainees do this. I can't think of anything specific at the moment, but if they say "I find that interesting, you can send them to me".*

Trainees should also try to establish a good relationship with other staff, as explained below:

*Supervisor no. 17: What also may help, especially in the beginning of their training year, is that, if a trainee does not have a full surgery, that they do not stay in their room, but go to the medical receptionist. Personal contact between receptionists and trainees. The receptionist will work harder for the trainee, if they have a good relationship.*

Trainees should also try to involve other staff in their training and learning, because this made staff more willing to allocate patients with a specific problem to them, as the following examples show:

*MR no. 2: Yes, If I know from this or that, hat she needs to discuss it at the educational day at the institute this week. I'll indeed try to plan it.*

### Attitude of team members
First encounters between patient and trainee are mainly arranged by supervisors, medical receptionists and nurse practitioners. However, team members sometimes struggled with achieving a balance between teaching the trainee and providing good patient care, which could affect the trainee patient mix. This is illustrated by the following conversation between two supervisors:

*Supervisor no. 2: I sometimes find it difficult. Recently, a woman phoned for me when I wasn't at the practice. She has recurrent lung cancer. The trainee took the call. Should I take over? should I contact the patient? Who should be in charge. I find it difficult.*

*Supervisor no. 5: What do you find difficult?*

*Supervisor no. 2: You don't want to let the patient go, you want the patient to know that you're there for her. On the other hand, she [the trainee] should also get the chance to do things.*

The feeling of being involved in the trainees' education was important. As shown above, the nurse practitioner and medical receptionist were more willing to take the trainee's learning objectives into consideration when allocating a patient if they felt involved in that trainees' education. However, not every team member was equally involved or wanted to be involved.

## Context

Our participants noticed that the context has also an important influence. For example, every practice has their own contextual factors such as the availability of a nurse practitioner, the organisation and location of a practice, the size of a team, the number of previous trainees and so on. An example of a contextual factor (having a walk-in surgery) that influenced trainee patient mix is explained by a supervisor:

*Supervisor no. 10: The walk-in surgery is always run by trainees in this practice. They can learn about minor acute problems but also make contact with patients, make an follow-up appointment with them. In this way you can build up your own patient list in no time. It's very satisfactory and works well.*

Context also seemed to affect strength of the doctor-patient relationship. If a general practice has only one doctor, then the doctor–patient relationship is probably strong and patients will be less willing to consult a trainee. This in contrast with larger practices where patients rotate between doctors, whereby patients have less strong ties with a single doctor.

*Trainee no. 15: I worked in a solo practice with a 64-year-old GP who had worked there for 34 years. He was THE doctor. So yes, it's probably good to gain experience in a health centre with eight GPs and three trainees, but whether it makes a difference to the patients? I don't know. In a solo practice the trusting relationship is very sure and resistant, certainly if the doctor has been there for long. Maybe somebody here has experience with this? To be in a health center with multiple doctors?*

*Trainee no. 16: Yes, I think that you [the doctor] will be more replaceable. When a patient always sees the same GP, than he always wants to go there. If there are at least 6 doctors it [the patient] would be more flexible, so I think it matters.*

## DISCUSSION

### Summary of main findings

The mix of patients GP trainees see is shaped by several factors. The characteristics of the health problem and the bond between trainee and patient are major determinants of this mix. Patients with complex problems tend to prefer their 'own doctor' because they trust him/her. It is generally recognised that a strong, trustful doctor–patient relationship is essential for the provision of high-quality care. Trainees cannot have such a relationship, especially not at the start of their training year in a new general practice. Our findings suggest that trainees and team members can take steps to surmount this lack of a trustful relationship. Team members can provide trainees with opportunities to see specific patients, and trainees can take specific steps to develop trustful doctor–patient relationships. Whether this happens depends on trainee behaviour, the attitudes of team members and the context of the general practice. Team members have

to trust the trainee before they are willing to allocate or transfer patients with complex medical conditions to the trainee. This is influenced by both practice culture and the trainees' behaviour. Trainees should invest in a good relationship with team members, involve them in their development and needs, and show a keenness to learn. Trainees should also invest time and energy in building a relationship with patients, by being proactive. This will help patients to trust them. In this way, trainees will generate their own list of patients with diverse medical problems of varying complexity and deliberately influence their own learning process in a positive way.

### Strengths and limitations

In this study, we took a broad point of view and incorporate the perspectives from multiple participants with different roles within the general practice, namely, the GP supervisor, the GP trainee, the medical receptionist and the nurse practitioner. Because of this variety in roles, we had to include a relatively large number of participants before we reached data saturation. Although we did not incorporate patients' perspectives, findings from previous studies on patients' perspectives indicated that their views overlapped with our findings.[8 28 29] The use of qualitative methods enabled us to look in depth at factors affecting the patient mix of GP trainees from different perspectives. The fact that our findings are based on the patient mix of a Dutch GP trainee could be regarded as a limitation for translating our findings to other settings. Though the importance of continuity of care, especially for patients with chronic conditions, is acknowledged overall,[28 30–32] contextual differences could affect the transferability of the influence this has on trainees' patient mix elsewhere. For example, Dutch GPs are rooted within communities which strengthens the bond between patients and doctors. In other places, doctors can have less or more bonding with their community, and therefore this might have less or more influence on GP trainees' patient mix. Moreover, it is less likely that our findings can be translated to a hospital setting, where the strong doctor–patient relationship seen in primary care is less likely to exist because doctor–patient contact there is more fleeting.

### Comparison with previous research and theory

A body of knowledge exists about the continuity of care, its relationship with the complexity of disease, and the high value patients and doctors place on it.[29 30 33] However, only a few studies have investigated this in relation to the mix of patients seen by GP trainees, and these studies have tended to take the patients' point of view as outcome.[8 15 29 34] Although we had a different perspective, our findings are in line with this research. Bonney *et al* found that older patients have a strong preference for their 'own' GP, and that their attitude towards the GP trainee has to be viewed within this perspective.[34] This finding is consistent with multiple cross-sectional studies from different countries.[10 28 29 35]

However, not only patients' preferences influenced the trainees' patient mix. We found that supervisors, nurse practitioners and medical receptionists also have their influence. This finding mirrors those of Bannister *et al*, who found that nurses and other team members in a hospital setting can hinder paediatric trainees in their learning opportunities.[6] Our finding of trainees as important influencers is consistent with the findings of multiple studies on how trainees learn. Among others, previous research has found that learner engagement is important for trainees to practice skills,[6] to create learning opportunities[36] and to create a more learning-oriented environment.[37 38]

Our findings are consistent with socio-cultural theories about learning.[39] The situated learning theory of Lave and Wenger emphasises the importance of participation in daily work for learning.[40] Learners are active participants, and through participation they become full members of a 'community of practice'. Newcomers evolve through engagement with their workplace and gain more responsibilities. We found that GP trainees need to engage in their communities: by actively participating and building work relationships, trainees will gain trust and gradually be given more responsibility. Trainees evolve then from a 'legitimate peripheral participator' into a legitimate member of the community. We found that trainees who made an effort and invested time in their work environment were important influencers of their learning and patient mix. This finding is consistent with earlier work by Billet and Bandura, who stated that learners have to engage to reach the full educational potential a workplace has to offer.[17 41]

### Implications for research and practice

The mix of patients seen by GP trainees should offer trainees the opportunities they need to develop competencies and to acquire skills. This study reveals that trainees' patient mix is affected by many entangled factors, but that trainees can take a central role in influencing their patient mix, both direct and indirectly. These findings have some implications for practice.

First, trainees should be aware that they can directly influence their own patient mix. Following our results, we recommend enlarging trainees' knowledge of the effects of the doctor-patient relationship and of ways in which they may speed up the bonding between themselves and their patients. For example, trainees can be taught to deliberately make social house visits, contact patients after their visit to the hospital or arrange that all follow-up visits will be with the trainee. Second, trainees should be encouraged to arrange with their supervisor or the nurse practitioner to transfer some patients to the trainee's care. However, trainees and their supervisors should be informed that with this transfer, a *transfer of trust* must also take place. Seeing a patient together with the trainee and supervisors/nurse practitioner or stressing the trainee's qualities may help ensure that patients receive the same quality of care with the trainee as they do with their former doctor.

Furthermore, trainees and supervisors should be informed that relationships between trainees and the team members of a practice indirectly influence trainees' patient mix. Therefore, trainees should be instructed to maintain good relationships by showing keenness, involving other team members in their learning and conversing with the medical receptionist about their experiences during consultations. Moreover, supervisors and nurse practitioners should be aware that their own perceptions and beliefs about patient care influence the patients a trainee sees. We recommend starting the training year with a discussion between the trainee, supervisor and other team members about these beliefs and the trainee's expectations.

The above-mentioned recommendations of this study, together with the results a subsequent study on patients' perspectives, will be used to design an educational intervention to enhance GP trainees' patient mix. This intervention study will lead to even more concrete and substantiated recommendations for trainees, supervisors and GP training programme directors.

**Acknowledgements** The authors wish to thank all participants who were willing to share their thoughts about trainees' patient mix, as well as the staff of the GP specialty training of the Amsterdam UMC (University of Amsterdam) who helped us with the organisation of the focus groups and interviews.

**Contributors** SdB, MV and NvD designed the study. SdB, SvR and MV performed the interviews and analysis with support from NvD and AK. All authors discussed the results. SdB wrote the manuscript with input from all authors.

**Funding** This work is funded by a grant from ZonMW (grant number: 839130001).

**Competing interests** None declared.

**Patient consent for publication** Not required.

**Ethics approval** Dutch Association for Medical Education (file number 630).

**Provenance and peer review** Not commissioned; externally peer reviewed.

**Data availability statement** Data are available upon reasonable request.

**ORCID iD**
Sarah de Bever http://orcid.org/0000-0002-5113-0396

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
