## [Reviewer comments · BMJ Open]

ARTICLE DETAILS

TITLE (PROVISIONAL)	Professionals' perspectives on factors affecting GP trainees' patient mix: Results from an interview and focus group study among professionals working in Dutch general practice.
AUTHORS	de Bever, Sarah; van Rhijn, Suzanne C; van Dijk, Nynke; Kramer, Anneke; Visser, Mechteld R.M.

VERSION 1 – REVIEW

REVIEWER	Roar Maagaard Center for Health Sciences Education, University of Aarhus. Denmark
REVIEW RETURNED	07-Jul-2019

GENERAL COMMENTS	Very relevant for GP trainers also outside the Netherlands. Results are not surprising in any ways - but here we have the scientific basis for the first time for the advice that many organisers of GP training give to trainers and trainees. The study is very well described - and I find the methods very relevant. There are some minor details: although I am not native English speaking I think there should be a final English proof reading. Examples: - from the summary: "...and interviewed with..." just "interviewed"- from the summary: "Using professionals this study..." - What do you mean? Do you mean something like this "By including different types of health care professionals this study...?"- in "Setting" you write "..., each trainee is supervised and coached by an experienced GP (supervisors)." - It must be "supervisor". Other remarks: in 2 places you lack information in the text - and then this information suddenly appears a little later - this is not optimal: 1. At the end of "Design" "...sampled on third year trainees,..." - at this stage only some of us know that you in the Netherlands have a 3 year GP training program, so we can understand why you did so - but some lines later you explain the NL-training program.2. At the end of "Design" you give an elaborated description of the training of GP supervisors, nurse practitioners and medical receptionists at "the GP Specialty Training Institute of the Amsterdam UMC, location AMC..." The reader would think: "and I could not care less...." (and so were my thoughts...) - but then it becomes obvious later that this is highly relevant as these groups are part of your study population! But please inform beforehand about this.
---

	A little remark: I think you should add in a bracket a brief editorial note on what "BIRADS-4" is - as I know that in some countries this classification is not used and GP trainers will not instantly know what this GP supervisor is talking about. In your final part: I think you really should emphasise more on how your findings can (and should) be used by trainers, trainees and the staff in GP-practices to optimise learning!
--	---

REVIEWER	Richard Hays James Cook University Australia
REVIEW RETURNED	07-Aug-2019

GENERAL COMMENTS	This is a well written paper that addresses an important issue in GP specialty training. Although I have no experience with GP training in the Netherlands, the results are credible and fit with my observation in other countries with similar training programs. Some methodological issues could be improved. First, I could not find a clear ethics approval statement, so this should be added. Second. I am not sure why the authors did so many interviews and focus groups. 73 participants is a lot for a qualitative study. Was this to ensure representation from a wide variety of practices? I also think that more could be made of the information and analysis to go beyond barriers. For example, is there a list of tips for training practices to facilitate the development of trust faster? Is this harder in solo practices? I know that many training practices deliberately allocate some patients with longer term problems to trainees, and some patients are cared for quite happily by a succession of trainees. Did suggestions like this emerge from the analysis? Overall, even as is, a useful contribution, although not really adding much that is new.
---

VERSION 1 – AUTHOR RESPONSE

Reviewer # 1		
Comment	Answer	Page
Comment 1: There are some minor details: although I am not native English speaking I think there should be a final English proof reading. Examples: - from the summary: "...and interviewed with..." just "interviewed"	Thank you for bringing the quality of our English to our attention. As mentioned in our accompanying letter, we have send those paragraphs lacking earlier language editing to a professional language editing service.	See track changes throughout the whole manuscript

- from the summary: "Using professionals this study..." - What do you mean? Do you mean something like this "By including different types of health care professionals this study..."? - in "Setting" you write "..., each trainee is supervised and coached by an experienced GP (supervisors)." - It must be "supervisor".		
Comment 3: 2 places you lack information in the text - and then this information suddenly appears a little later - this is not optimal: 3.1. At the end of "Design" "...sampled on third year trainees,..." - at this stage only some of us know that you in the Netherlands have a 3 year GP training program, so we can understand why you did so - but some lines later you explain the NL-training program. 3.2. At the end of "Design" you give an elaborated description of the training of GP supervisors, nurse practitioners and medical receptionists at "the GP Specialty Training Institute of the Amsterdam UMC, location AMC..." The reader would think: "and I	Comment 3.1: We agree that this may be confusing for readers who are not familiar with the Dutch GP training program. Therefore, we moved the sentence about the sampling on third year trainees to the 'Participants' section, and changed 'third year trainees' into 'final year trainees' Comment 3.2: To improve the logical order in our manuscript, we start with the 'Participants' section, followed by the 'Setting' section in which we explain the training of the different participants.	Comment 3.1. - page 7 paragraph Participants, yellow text Comment 3.2 - Page 7, 8, 9, yellow text
could not care less...." (and so were my thoughts...) - but then it becomes obvious later that this is highly relevant as these groups are part of your study population! But please inform beforehand about this.		

Comment 4: A little remark: I think you should add in a bracket a brief editorial note on what "BIRADS-4" is - as I know that in some countries this classification is not used and GP trainers will not instantly know what this GP supervisor is talking about.	Thank you for noticing our use of terms that are not internationally known. As you suggested, we inserted the following editorial note, explaining BIRADS. red: Breast Imaging Reporting and Data System (BIRADS). A standardized reporting system for mammography, BIRADS 4 indicates that a suspicious node has been seen	Page 15, yellow text
Comment 6: I think you really should emphasise more on how your findings can (and should) be used by trainers, trainees and the staff in GP practices to optimise learning!	Both you and the other reviewer addressed the lack of suggestions on how to use our findings. We are grateful to provide more concrete recommendations. We added those to the paragraph 'Implications for practice and research' in the discussion.	Page 22 & 23, yellow text

Reviewer # 2		
Comment 2: Some methodological issues could be improved. First, I could not find a clear ethics approval statement, so this should be added.	We agree that our ethical statement was somewhat hidden. In our revised document we included a separate Ethic Approval paragraph in our Method section	Page 11, yellow text
Commet 3: Second. I am not sure why the authors did so many interviews and focus groups. 73 participants is a lot for a qualitative study. Was this to ensure representation from a wide variety of practices?	We agree that 73 participants is a lot for a qualitative study. However, we included four different 'types' of participants: GP trainees, GP supervisors, medical receptionists and nurse practitioners. To reach data saturation for each perspective,	Page 19 & 20, yellow text

	we needed to include enough participants representing each role. This was especially pronounced for the trainees, since we sampled them on level of training (first, second and third year). We added an explanation for the large number of participants in our discussion section.	
Comment 4: I also think that more could be made of the information and analysis to go beyond barriers. For example, is there a list of tips for training practices to facilitate the development of trust faster? Is this harder in solo practices? I know that many training practices deliberately allocate some patients with longer term problems to trainees, and some patients are cared for quite happily by a succession of trainees. Did suggestions like this emerge from the analysis? Overall, even as is, a	As mentioned above, you and the first reviewer both commented on our lack of concrete tips from our findings. We would like to thank you for the opportunity to make more concrete recommendations based on our results. Though we cannot provide a list of tips specifically related to, for example, practice size, we did incorporate recommendations to train trainees to take charge of their patient mix in the	Page 22 & 23, yellow text
useful contribution, although not really adding much that is new.	discussion section. We believe this is a useful method to influence trainees' patient mix and we are currently working on a study investigating such an intervention.	

VERSION 2 – REVIEW

REVIEWER	Roar Maagaard CESU (Center for Health Sciences Education) University of Aarhus Denmark
REVIEW RETURNED	13-Oct-2019

GENERAL COMMENTS	After revision I find the abstract absolutely ready for publishing. (I found a few misspellings - but apart from that nothing).
---

REVIEWER	Richard Hays
-----------------	--------------

	James Cook University Australia
REVIEW RETURNED	20-Sep-2019

GENERAL COMMENTS	The paper presents qualitative research into an important question in GP training. Can the diversity of patient encounters be increased faster? The study addresses this question through the eyes of participants, including patients and practice receptionists. These findings seem appropriate and fit with experience. My only concern is that the analysis is not provided in detail, so how the authors reached their findings is not really clear. It would be better if the thematic analysis were summarised in a table or an appendix to better support the findings. There are also some minor contextual differences between general practice in the Netherlands and elsewhere, so some of the discussion may be a little less useful outside of the Netherlands. That does not matter in qualitative research, but perhaps this could be stated more strongly?
---

VERSION 2 – AUTHOR RESPONSE

Reviewer # 1		
Comment	Answer	Page
Comment 1: My only concern is that the analysis is not provided in detail, so how the authors reached their findings is not really clear. It would be better if the thematic analysis were summarised in a table or an appendix to better support the findings	Thank you for pointing this out. We found it a bit difficult to fully understand this request. We have added an appendix in which we provide a table of our thematic analysis in which we summarize our main themes and their subthemes. We modelled this to table to the one in Gupta et al (Med Ed 2014). We hope this fulfills your request. If not, please let us know in more detail what is intended with a summary of the thematic analysis.	Appendix I and page 12, yellow text
Comment 2: There are also some minor contextual differences between general practice in the Netherlands and elsewhere, so some of the discussion may be a little	We agree that context can be quite different between countries. Therefore we included a comment that transferability of our findings	Page 21, yellow text

less useful outside of the Netherlands. That does not matter in qualitative research, but perhaps his could be stated more strongly?	might be affected by these differences.	
--	---	--

VERSION 3 – REVIEW

REVIEWER	Richard Hays James Cook University Australia
REVIEW RETURNED	19-Nov-2019

GENERAL COMMENTS	The reviewer completed the checklist but made no further comments.
--